# QADYNAMICS: Training Dynamics-Driven Synthetic QA Diagnostic for Zero-Shot Commonsense Question Answering

**Haochen Shi**[*], **Weiqi Wang**[*], **Tianqing Fang, Baixuan Xu, Wenxuan Ding,**
**Xin Liu, Yangqiu Song**

Department of Computer Science and Engineering, HKUST, Hong Kong SAR, China
hshiah@connect.ust.hk, {wwangbw, tfangaa, yqsong}@cse.ust.hk

## Abstract

Zero-shot commonsense Question-Answering (QA) requires models to reason about general situations beyond specific benchmarks. State-of-the-art approaches fine-tune language models on QA pairs constructed from Common-Sense Knowledge Bases (CSKBs) to equip the models with more commonsense knowledge in a QA context. However, current QA synthesis protocols may introduce noise from the CSKBs and generate ungrammatical questions and false negative options, which impede the model's ability to generalize. To address these issues, we propose QADYNAMICS, a training dynamics-driven framework for QA diagnostics and refinement. Our approach analyzes the training dynamics of each QA pair at both the question level and option level, discarding machine-detectable artifacts by removing uninformative QA pairs and mislabeled or false-negative options. Extensive experiments demonstrate the effectiveness of our approach, which outperforms all baselines while using only 33% of the synthetic data, even including LLMs such as ChatGPT. Moreover, expert evaluations confirm that our framework significantly improves the quality of QA synthesis. Our code and model checkpoints are available at https://github.com/HKUST-KnowComp/QaDynamics.

## 1 Introduction

The advent of various commonsense Question-Answering (QA) benchmarks (Talmor et al., 2021; Huang et al., 2019) has demonstrated that Pre-Trained Language Models (PTLMs) (Devlin et al., 2019; Lan et al., 2020) can achieve extraordinary performances when fine-tuned on these benchmarks. However, these neural systems have been criticized for only learning surface-level correlations and lacking general semantic reasoning abilities, which often require implicit commonsense

---

[*] Equal Contribution

knowledge (Branco et al., 2021; Zhou et al., 2021). To reliably assess the resilience of QA models across diverse domains, the zero-shot commonsense QA task has been proposed to evaluate the generalizable reasoning ability of a QA model (Li et al., 2020; Shwartz et al., 2020) without supervision signals from any QA benchmarks.

Ma et al. (2021) introduced a technique for tackling this task by fine-tuning a PTLM on QA pairs synthesized from knowledge triples in Common-Sense Knowledge Bases (CSKBs). The head and relation of a triple were transformed into a question using natural language templates, with the tail serving as the answer. Distractors, or negative examples, were tails from triples sampled from the same CSKB using pre-defined strategies, such as keyword or embedding proximity filtering. However, the primary obstacle hindering further progress in this method is the quality of the synthetic QA dataset. This issue arises because manually-curated CSKBs often contain subtle but strong annotation artifacts (Zellers et al., 2019; Sakaguchi et al., 2021), which could provide *easy* back-doors for the model to perform exceptionally well on synthetic test sets but fail to generalize on held-out QA benchmarks. Additionally, the current QA synthesis process results in a significant number of ungrammatical questions, and the negative sampling strategy used to create distractors is not entirely effective in preventing false-negative options, as evidenced by Ma et al. (2021).

Despite the existence of dataset filtering algorithms, such as adversarial filtering (Zellers et al., 2018) for negative option selection, they have been shown to be less effective compared to random selection baselines (Ma et al., 2021). This is because they only focus on model uncertainty in the final predictions, which is not effective enough for synthetic data that contains a plethora of noise and imbalanced examples (Appx. §A.1).

Instead of leveraging data filtering based on

model uncertainty in the final predictions, we draw inspiration from Swayamdipta et al. (2020) and employ *training dynamics* as a more precise indicator that studies instance learnability across all training steps. While the vanilla training dynamics regard each data instance as a whole without considering the learnability of each option or choice, we propose QADYNAMICS, a training dynamics-driven framework for synthetic QA diagnostic and refinement that favors choice-level diagnosis. Specifically, our approach proposes a novel schema that offers greater flexibility in deriving the training dynamics of multiple-choice QA with an arbitrary number of options, thus accommodating the varying number of choices in different commonsense QA benchmarks. QADYNAMICS then analyzes the training dynamics of each option, greedily drops the *easy* distractor to reduce the impact of CSKB artifacts, and eliminates QA pairs containing mislabeled or false-negative options according to the confidence gap between all options (§3). Extensive experiments showcase the efficacy and data efficiency of our proposed framework, surpassing all previous zero-shot CSQA baselines while only leveraging 33% of training data and even outperforming GPT3.5 (Ouyang et al., 2022) and ChatGPT (§4.4). Further expert evaluations confirm the effectiveness of our proposed method in enhancing the quality of the synthetic QA set (§4.5).

## 2 Related Works

### 2.1 Zero-shot Commonsense QA

The task of zero-shot commonsense QA requires a QA model to perform generalizable QA towards commonsense questions from held-out benchmarks whose training data is inaccessible to the model. Existing approaches either leverage off-the-shelf language models in an unsupervised manner to unlock their commonsense capability with inference time mechanisms, such as self-talk (Shwartz et al., 2020), cloze translation (Dou and Peng, 2022), and dynamic graph reasoning (Bosselut et al., 2021), or inject commonsense knowledge into PLMs by fine-tuning them on synthetic QA pairs constructed from CSKBs (Ma et al., 2021; Kim et al., 2022; Wang et al., 2023a; Zhang et al., 2022). While unsupervised approaches only achieve satisfactory performances, existing works following the fine-tuning regime have shown exceptional performances on various commonsense QA benchmarks. However, fine-tuning heavily relies on the quality

of training data, which is subject to limitations in both the knowledge quality and coverage in the CSKBs and the protocol for synthesizing them into QA pairs. Both of these are restricted by specific limitations, as discussed in §1.

### 2.2 Dataset Diagnostic

Diagnosing individual data instances within a large dataset has long been an important aspect of machine learning for NLP (Deng et al., 2023). Various data attribution methods have been proposed to retrieve training instances that may have led to a particular prediction (Pezeshkpour et al., 2021; Xie et al., 2023). Building on this, Pezeshkpour et al. (2022) proposed a method to efficiently detect dataset artifacts in the training data using data attribution methods when a challenging validation set is available. While these methods focus on the impact of individual instances on specific predictions, more generalized and precise dataset diagnostic approaches have also been proposed (Swayamdipta et al., 2020; Ethayarajh et al., 2022). These approaches aim to understand the difficulty of learning specific instances and can detect annotation artifacts and perform automatic data corrections, such as mislabeling detection. However, none of these methods explicitly consider QA benchmarks where each QA pair contains more than one piece of knowledge. To effectively evaluate the attribution of a QA pair, it is necessary to consider all possible options for fair consideration.

## 3 QADYNAMICS

This section outlines our proposed framework, QADYNAMICS, which consists of four steps: (1) Calculate the training dynamics for each option in a QA pair. (2) Refine the QA pair by eliminating the *easy* distractor. (3) Filter out QA pairs that may be mislabeled or contain false-negative distractors. (4) Train the model using marginal ranking loss.

### 3.1 Preliminary

We follow the pipeline and task definition formulated by Ma et al. (2021) to study the zero-shot commonsense QA task. Formally, denote a CSKB as $D = \{(h, r, t)|h \in H, r \in R, t \in T\}$, where $H, R, T$ are the sets of heads, relations, and tails. Every triple in $D$ is transformed into a $(Q, A)$ pair, where $Q$ is a question constructed using $(h, r)$ with natural language templates and $A = \{A_1, A_2, \ldots, A_m\}$ is the corresponding set

of options containing $m$ choices. Specifically, $t$ is used as the ground-truth answer $A_1$, and other distractors are tails from $m-1$ triples sampled using keyword overlap filtering. The objective is to obtain a QA model $\theta$ from the synthetic QA sets $D^Q = \{(Q_i, A_i) | (h_i, r_i, t_i) \in D\}$ and test $\theta$ on held-out QA benchmarks.

## 3.2 Training Dynamics of QA Pairs

Following Ma et al. (2021), the QA model is trained through fine-tuning a pre-trained masked language model. For a given $(Q, A)$ pair, $Q$ is concatenated with every option $A_i \in A$ first to obtain the input sequence $T_i$. We then repeatedly mask out a token in $T_i$ at one time and calculate the model's masked loss. The logit score of $T_i$ with $n$ tokens is calculated by:

$$\mathcal{S}(T_i) = -\frac{1}{n} \sum_{i=1}^{n} \log P(t_i | t_1, ..., t_{i-1}, t_{i+1}, ..., t_n) \quad (1)$$

Intuitively, the option with the lowest logit score will be selected as the answer. Based on this, we introduce our proposed schema for calculating the training dynamics of $(Q, A)$ at both the pair level and option level. Following Swayamdipta et al. (2020), we train a QA model $\theta'$ on $D^Q$ and save $E$ checkpoints $\{\theta'_1, \theta'_2, \ldots, \theta'_E\}$ along the training process. At checkpoint $\theta'_e$, denote $T_j$ as the input sequence with the second lowest logit of distractors among those containing a distractor, the model's *confidence* of $T_1$ (concatenation of $Q$ and $A_1$) being correct is:

$$P(\theta'_e, T_1) = \frac{\exp(-S(T_1))}{\exp(-S(T_1)) + \exp(-S(T_j))} \quad (2)$$

Similarly, the *confidence* of a distractor's input sequence $T_i$ being wrong is defined as:

$$P(\theta'_e, T_i) = 1 - \frac{\exp(-S(T_i))}{\sum_{k=1}^{m} \exp(-S(T_k))} \quad (3)$$

Based on the *confidences* of all options, we formulate the *confidence* of a $(Q, A)$ pair as:

$$P(\theta'_e, Q, A) = \frac{1}{m} \sum_{k=2}^{m} (P(\theta'_e, T_1) + P(\theta'_e, T_k) - 1) \quad (4)$$

Finally, following Swayamdipta et al. (2020), we derive scores for each option and QA pair at each of the $E$ checkpoints using the equations defined above. The final *confidence* and *variability* scores are obtained by calculating the average and standard deviation of these scores across $E$ checkpoints (more detailed explanations in Appx. §A.1).

## 3.3 Option Selection

To reduce any artifacts present in the synthetic QA set that may have originated from the CSKBs, we adopt a similar approach to AFLite (Bras et al., 2020) and remove negative knowledge that the model can easily identify. We achieve this by discarding one distractor with the highest *confidence* score, which indicates that the model may be susceptible to exploiting potential biases and consistently assigns a high score to this option. We then concatenate the modified option set $A'$, containing the original ground-truth answer and $m-2$ distractors that are more challenging to distinguish, with the original question $Q$ to yield a more challenging $(Q, A')$ pair. Such an option level selection strategy is termed as *Difficult Choice*.

## 3.4 QA Pair Selection

Next, to improve the quality of the synthetic QA set, we remove poor-quality QA pairs that contain the following two types of options:

**Mislabeled Ground-Truth Option.** We remove the QA pairs whose correct answer is associated with very low confidence, indicating potentially being *mislabaled* (Swayamdipta et al., 2020).

**False Negative Distractor.** We remove QA pairs where the difference in confidence score between the ground-truth answer and the distractor with the highest confidence score is insignificant. This indicates the potential for a false negative.

## 3.5 Model Training

Finally, we fine-tune $\theta$ on our cleaned synthetic QA set using marginal ranking loss. With the score of each option defined in Equation (1), the marginal ranking loss, with $\eta$ being the margin, is:

$$\mathcal{L} = \frac{1}{m-1} \sum_{i=2}^{m-1} max(0, \eta - \mathcal{S}(T_1) + \mathcal{S}(T_i)) \quad (5)$$

# 4 Experiments

## 4.1 Datasets

Following Ma et al. (2021), we leverage the combination of five CSKBs, including ATOMIC (Sap et al., 2019a), ConceptNet (Speer et al., 2017), WordNet (Miller, 1995), VisualGenome (Krishna et al., 2017), and WikiData (Vrandecic and Krötzsch, 2014), as our source commonsense knowledge repository $D$. We then use the validation split of five commonsense QA benchmarks, including AductiveNLI (aNLI; Nie et al.,

Table 1:

| Data Selection Strategy | aNLI | CSQA | PIQA | SIQA | WG | Avg. |
|---|---|---|---|---|---|---|
| DeBERTa-v3-Large (Zero-shot; He et al., 2023) | 59.9 | 25.4 | 44.8 | 47.8 | 50.3 | 45.6 |
| Random 33% | 77.3 | 68.5 | 79.5 | 62.5 | 75.5 | 72.7 |
| Random 50% | 75.8 | 68.8 | 79.7 | 60.3 | 64.2 | 69.8 |
| Random 66% | 77.3 | 66.8 | 78.5 | 64.0 | 75.4 | 72.4 |
| Total 100% (Ma et al., 2021) | 78.7 | 68.5 | 79.1 | 63.5 | 75.4 | 73.0 |
| Total 100% (AF-Lite) (Ma et al., 2021) | 79.5 | 65.7 | 75.6 | 55.7 | 75.1 | 70.3 |
| **Large Language Models** | | | | | | |
| GPT-3.5 (`text-davinci-003`) | 61.8 | 68.9 | 67.8 | 68.0 | 60.7 | 65.4 |
| ChatGPT (`gpt-3.5-turbo`) | 69.3 | **74.5** | 75.1 | **69.5** | 62.8 | 70.2 |
| **TRAININGDYNAMICS – DeBERTa-v3-Large** *435M* | | | | | | |
| *66% train* Easy-to-learn | 75.9 | 67.9 | 75.6 | 62.9 | 73.2 | 71.1 |
| Ambiguous | 80.0 | 69.1 | 80.3 | 63.2 | 78.4 | 74.2 |
| Hard-to-learn | 79.7 | 69.0 | 78.9 | 63.8 | 77.7 | 73.8 |
| Hard-to-learn *Mislabeled.* | 80.3 | 70.3 | 79.4 | 63.9 | 77.4 | 74.3 |
| Hard-to-learn *False-Neg.* | 80.9 | 69.5 | 79.7 | 64.2 | 77.4 | 74.3 |
| Hard-to-learn Mixed Strategy | 80.7 | 70.1 | 79.6 | 63.8 | 77.3 | 74.4 |
| *33% train* Easy-to-learn | 75.1 | 67.9 | 78.0 | 65.0 | 74.8 | 72.2 |
| Ambiguous | 80.9 | 70.1 | 78.4 | 62.5 | **79.6** | 74.3 |
| Hard-to-learn | 80.9 | 67.5 | 79.4 | 62.6 | 76.9 | 73.5 |
| Hard-to-learn *Mislabeled.* | 79.5 | 70.1 | 78.2 | 62.0 | 79.2 | 73.8 |
| Hard-to-learn *False-Neg.* | 80.7 | 70.4 | 78.5 | 66.5 | 77.3 | 74.7 |
| Hard-to-learn Mixed Strategy | **82.3** | 70.9 | 78.6 | 64.5 | 78.0 | 74.9 |
| **QADYNAMICS (Ours) – DeBERTa-v3-Large** *435M* | | | | | | |
| *66% train* Easy Choice | 75.2 | 69.5 | 77.2 | 62.7 | 71.7 | 71.3 |
| Difficult Choice | 80.4 | 68.8 | 79.0 | 64.0 | 76.9 | 73.8 |
| Difficult Choice – *Mislabeled.* | 80.0 | 70.1 | 79.4 | 63.8 | 79.4 | 74.2 |
| Difficult Choice – *False-Neg.* | 79.0 | 70.8 | 79.8 | 63.3 | 78.9 | 74.4 |
| Difficult Choice – Mixed Strategy | 80.0 | 70.7 | 80.4 | 65.1 | 78.6 | 75.0 |
| *33% train* Difficult Choice – Hard-to-learn without strategy | 79.7 | 71.5 | 79.9 | 65.4 | 76.1 | 74.5 |
| Difficult Choice – Hard-to-learn *Mislabeled.* | **82.3** | 72.2 | 79.8 | 63.3 | 79.0 | 75.3 |
| Difficult Choice – Hard-to-learn *False-Neg.* | 81.9 | 70.4 | 79.7 | 66.9 | 79.0 | 75.6 |
| Difficult Choice – Hard-to-learn Mixed Strategy | **82.3** | 71.6 | **81.2** | 65.6 | 79.1 | **76.0** |
| **Supervised Learning & Human Performance** | | | | | | |
| DeBERTa-v3-L (Supervised) | 89.0 | 82.1 | 84.5 | 80.1 | 84.1 | 84.0 |
| Human Performance | 91.4 | 88.9 | 94.9 | 86.9 | 94.1 | 91.2 |

Table 1: Zero-shot commonsense QA evaluation results on five benchmarks (Accuracy %). All experiments employ DeBERTa-v3-Large (He et al., 2023) as the backbone. The best performances are **bold-faced**, and the second-best ones are underlined. "Mislabeled." refers to removing QA pairs whose ground-truth answer is mislabeled, and "False-Neg." refers to removing QA pairs containing false-negative distractors (§3.4). "Mixed Strategy" indicates iteratively applying both measures above to eliminate poor-quality QA pairs.

2020), CommonsenseQA (CSQA; Talmor et al., 2019), PhysicalIQA (PIQA; Bisk et al., 2020), SocialIQA (SIQA; Sap et al., 2019b), and Wino-Grande (WG; Sakaguchi et al., 2021), for evaluation. More statistics are shown in Tab. 4.

### 4.2 Dataset Statistics

In our method, we set a threshold to filter out mislabeled and false negative data from the entire dataset. Intuitively, it is essential to establish the accuracy and reliability of the data before proceeding with any further division or analysis. The threshold is decided based on rough observations of QAdynamic distributions, emphasizing the balance between quantity and quality.

The specific statistics are shown in Tab. 2. As mentioned by Ma et al. (2021), the human accuracy on ATOMIC and CWWV synthetic data is 78.0 and

| | Mislabeled | False-Neg. | Mixed Strategy | Total |
|---|---|---|---|---|
| Data size | 6465 | 26320 | 32875 | 345775 |
| Ratio | 0.94% | 3.80% | 4.74% | 100% |

Table 2: Statistics of the number of QA pairs that are dropped by each strategy.

80.7, respectively. The data discovered automatically by our strategy is 4.74% of total data, which is close to 25% of the poor-quality or grammatically wrong data. Most of them are located in the low-confidence areas, indicating our framework's contribution towards purifying the low-quality data.

### 4.3 Experiment Setup and Baselines

We use accuracy as the evaluation metric. To derive the QADYNAMICS of the synthetic QA entries, we use RoBERTa-large (Liu et al., 2019) as the backbone of $\theta^I$, and for our final QA model $\theta$, we use

DeBERTa-v3-large (He et al., 2023). Our choice to utilize different models is because RoBERTa-large results in faster training and inference speed, and intuitively, it is challenging to expect a model to learn from data that is itself difficult to learn. We compare our results with several baselines to demonstrate the effectiveness of our training dynamics-driven data selection. First, we include those using 33%, 66%, and 100% synthetic QA pairs that are generated using keyword filtering or AFLite for distractor selection. We also report the performance of Large Language Models (LLMs), including GPT3.5 (Brown et al., 2020; Ouyang et al., 2022) and ChatGPT (OpenAI, 2022), as competitive baselines. To provide a fair comparison, we compare our framework with the original training dynamics-based data selection (Swayamdipta et al., 2020) with equal amounts of training data (33% and 66%). We select QA pairs that are *easy-to-learn*, *ambiguous*, and *hard-to-learn*, according to their confidence and variability distribution, and perform mislabeled correction on the *hard-to-learn* data, as done by Swayamdipta et al. (2020). For our framework, we utilize our proposed *Difficult Choice* selection (§3.3) with a combination of QA pair selection strategies (§3.4). Furthermore, we operate our framework on 50% of total QA pairs that have low confidence to show the effectiveness of our framework on *hard-to-learn* data. More explanations are provided in Appx. §A.1.

## 4.4 Results

The main results are shown in Tab. 1. Consistent with Swayamdipta et al. (2020), we observe that training the model with *ambiguous* and *hard-to-learn* data leads to the largest benefit in the baselines, outperforming both random data selection and LLMs. Mislabeled correction on *hard-to-learn* data also has a positive impact, indicating that the synthetic QA entries indeed contain such errors. Our best system, trained on *hard-to-learn* QA entries and applying all option and QA selection strategies, achieves state-of-the-art results by significantly outperforming all baselines on most benchmarks. It outperforms the best baseline (*Ambiguous*) by 1.7% in terms of average accuracy and surpasses LLMs by 5.8%. This demonstrates that dropping easy distractors to make the training set more difficult contributes to a more generalizable QA model, and leveraging QA selection strategies (§3.4) also has a positive impact, demonstrating the

| Data Selection Strategy | Plau.↑ | Mis.↓ | F.Neg↓ |
|---|---|---|---|
| Total Data | 80.0 | 18.0 | 30.0 |
| Hard-to-learn | 67.0 | 26.0 | 32.0 |
| After Removing Mislabeled. | **81.0** | **15.0** | 35.0 |
| Dropped by Mislabeled. | 18.0 | 70.0 | 35.0 |
| After Removing False-Neg. | 68.0 | 25.0 | **22.0** |
| Dropped by False-Neg. | 55.0 | 36.0 | 52.0 |
| After Applying Mixed Strategy | 76.0 | 17.0 | 25.0 |
| Dropped by Mixed Strategy | 54.0 | 43.0 | 45.0 |

Table 3: Expert evaluation results (%) on QA pairs selected using different combinations of strategies, which correspond to those defined in Tab. 1. Plau., Mis., and F.Neg refer to the ratio of QA pairs being plausible, containing mislabeled QA options, and containing false-negative distractors.

reliability of combining all proposed techniques in QADYNAMICS. Ablation studies are provided in Appx. §A.3.

## 4.5 The Effect of Option Selection

To verify the effectiveness of our option selections (§3.3), we recruit five graduate students specializing in machine commonsense to evaluate the quality of 100 randomly sampled synthetic QA pairs selected by various strategies. The experts are asked to annotate whether a QA pair is plausible (question and answer forms a plausible commonsense knowledge), mislabeled (the ground-truth answer is incorrect), or contains any false-negative distractor (the distractor is semantically correct). Our results, presented in Tab. 3, are consistent with the targeting effect of both strategies, which successfully reduces the ratio of mislabeled examples and false-negative examples. We also observe that jointly adopting both strategies benefits all three metrics, which positively supports the success of our best system in §4.4. Case studies are provided in Appx. §B.

## 5 Conclusions

In this paper, we propose QADYNAMICS, a training dynamics-empowered framework for data-efficient zero-shot commonsense QA that jointly considers the learning difficulty at both the QA and option levels. Our framework, on average, achieves state-of-the-art performance by surpassing large language models and all baselines significantly with only 33% of training data. Further expert evaluations showcase that our proposed method effectively eliminates poor-quality QA entries in the synthetic dataset.

## Limitations

The major limitation of QADYNAMICS is that our improved schema for assessing the training dynamics of a QA pair requires at least three options. This is because we consider all distractors when evaluating the *confidence* of the ground-truth answer and the entire QA pair, requiring more than one distractor to ensure precision. While most synthetic QA sets satisfy this requirement, there are also QA benchmarks that only have two options per question, such as WinoGrande (Sakaguchi et al., 2021) and aNLI (Nie et al., 2020). In such cases, the original training dynamics proposed by Swayamdipta et al. (2020) can be properly leveraged to deal with binary questions. We believe that this limitation is minor compared with the data-cleaning effect of QADYNAMICS.

## Ethics Statement

This paper uses datasets and benchmarks solely for research purposes, consistent with their intended usage. The expert student annotators recruited for this study were well-trained and agreed to participate voluntarily without receiving any payment. Since QADYNAMICS is a QA model and not a generative model, it does not yield additional biased content. Therefore, to the best of our knowledge, this paper does not involve any ethical concerns.

## Acknowledgements

The authors would like to thank the anonymous reviewers for their constructive comments. The authors of this paper were supported by the NSFC Fund (U20B2053) from the NSFC of China, the RIF (R6020-19 and R6021-20), and the GRF (16211520 and 16205322) from RGC of Hong Kong. We also thank the support from the UGC Research Matching Grants (RMGS20EG01-D, RMGS20CR11, RMGS20CR12, RMGS20EG19, RMGS20EG21, RMGS23CR05, RMGS23EG08).

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

# Appendices

# A  Additional Explanations and Experiments

## A.1  Motivations and Definitions of QADYNAMICS

Swayamdipta et al. (2020) proposed training dynamics, a novel model-based offline data selection method for NLP classification tasks. It obtained the data statistics during the training process and proposed two measures, *confidence* and *variability*, to assess the difficulty of a particular instance for the model to learn.

Fomally, consider a training dataset of size N, $D = \{(\boldsymbol{x}, y)_i\}_{i=1}^N$. The confidence of an instance $(x_i, y_i)$ is defined as:

$$\hat{\mu}_i = \frac{1}{E} \sum_{e=1}^{E} p_{\theta^{(e)}}(y_i \mid x_i) \qquad (6)$$

where $p_{\theta^e}$ denotes the probability predicted by the model parameters $\theta^e$ at the end of $e^{th}$ epoch.

And the variability measures the stability of $p_{\theta^{(e)}}(y_i|x_i)$, which is defined as:

$$\hat{\sigma}_i = \sqrt{\frac{\sum_{e=1}^{E} \left(p_{\boldsymbol{\theta}^{(e)}}\left(y_i \mid \boldsymbol{x}_i\right) - \hat{\mu}_i\right)^2}{E}} \qquad (7)$$

Given the definition of *confidence* and *variability* above, following Swayamdipta et al. (2020), the training data can be distinguished into three distinct regions, *easy-to-learn*, *ambiguous*, and *hard-to-learn*, respectively corresponding to high confidence, high variance, and low confidence. To obtain the subsets of *easy-to-learn* and *hard-to-learn*, we sort the dataset by confidence and take a certain percentage of data with the highest or lowest confidence (which, in our experiments, can be 33% and 66%). Similar to *Easy-to-learn*, to obtain *Ambiguous*, we take data with the highest variance. As stated by Swayamdipta et al. (2020), the *ambiguous* and *hard-to-learn* regions lead to the largest benefits on out-of-distribution performance.

However, when training with a QA dataset that includes $m$ distractors ($m > 2$), the confidence of the correct choice tends to be underestimated due to the larger number of distractors compared to the correct choice. To illustrate, given five options, where the logits associated are as follows: -1, -3, -3, -3, -3. Among these logits, the logit of the ground

truth option is -1. In this case, the confidence assigned to the correct choice is 0.65, while the confidence level assigned to the distractors is uniformly 0.91, indicating the logits of the ground-truth answer is relatively lower. Moreover, a model in the early training stage may make random guesses toward the answer, with a probability of approximately $1/m$ for each candidate. The probability of correct choice should gradually approach 1, resulting in lower confidence in the ground-truth answer than the distractors. Additionally, affected by the false-negative distractors, the confidence in the correct option may be underestimated relative to the true value. To alleviate the effect of data imbalance and false negative choice, as defined in Equation (2), we compute the confidence by only comparing the logit score of the correct answer with the logit score of the easier distractor, which is less likely to be a false negative. To verify the above statements, we compute the density of the difference between the logits of ground-truth answers and distractors. As shown in figure Fig. 1, compared to Softmax, our method has a higher density in the vicinity of 0, indicating the difference between logit scores is decreased. It can be stated that our method narrows the distribution gap between positive and negative options. With the above definition, high confidence in correct choice indicates a high probability of being chosen, and low confidence may indicate the question is mislabeled.

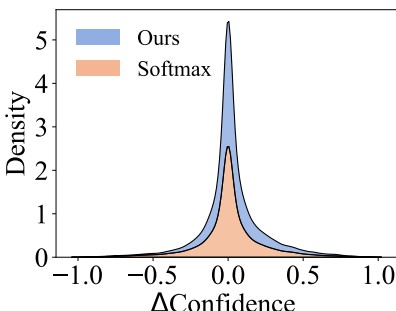

Figure 1: The density of difference between the confidence of ground-truth answer and distractors.

Unlike natural language inference data, which is used in Dataset Cartography (Swayamdipta et al., 2020), when evaluating confidence for a given QA pair, we should consider the confidence of all available options. As a result, we define the confidence of a QA pair as Equation (4). A higher confidence level for a QA pair indicates that the positive choice

| | aNLI | CSQA | PIQA | SIQA | WG |
|---|---|---|---|---|---|
| Question Numbers | 1532 | 1221 | 1838 | 1954 | 1267 |
| Choice Numbers | 2 | 5 | 2 | 3 | 2 |

Table 4: Statistics of the validation set of each benchmark.

is more likely to be selected, while the distractors are less likely to be chosen. To implement the *Difficult Choice* selection method, we remove one distractor with higher confidence. When we apply this method to the synthetic QA dataset, which has three candidates, 33% of the data is discarded, resulting in 66% of total data. For *Hard-to-learn* subset containing 50% of the total data, the amount of data becomes 33%.

As stated by Ma et al. (2021), the synthetic QA dataset includes ungrammatical questions as well as false negative distractors that appear plausible within the QA pair. Moreover, Dataset Cartography (Swayamdipta et al., 2020) suggests that confidence can also be used as a flag to identify mislabeled instances in the dataset. Thus, to deal with these two issues, we suggest two strategies: *Mislabeled.* removal and *False-Neg.* removal (§3.4). *Mislabeled.* involves excluding QA pairs with a low-confidence ground truth answer, while *False-Neg.* involves excluding QA pairs with correct answers and distractors with similar logits.

### A.2 Implementation Details

In this section, we introduce the implementations of our system. For hyperparameter tuning, following Ma et al. (2021), we set batch size 32, max sequence length 128, weight decay 0.01, warm-up proportion 0.05. We use an AdamW optimizer (Loshchilov and Hutter, 2019) with the learning rate set to 5e-6 in all experiments. We evaluate our models on the validation set of synthetic datasets every 1000 steps and save the one with the highest validation accuracy. Each experiment is repeated with different random seeds three times, and the average performance is reported. For computing resources, all of our experiments are conducted on 4 NVIDIA RTX A6000 GPUs, each with 48G memory. Our code for zero-shot commonsense QA is mainly based on the code repository provided by Ma et al. (2021), and all of the pre-trained language models are from the Huggingface Transformers Library (Wolf et al., 2020).

| Models | aNLI | CSQA | PIQA | SIQA | WG |
|---|---|---|---|---|---|
| QADYNAMICS | 82.3 | 71.6 | 81.2 | 65.6 | 79.1 |
| ⋄ w/o DC | 80.4 | 71.3 | 79.8 | 64.9 | 78.9 |
| ⋄ w/o *Mislabeled.* | 81.9 | 70.4 | 79.7 | 66.9 | 79.0 |
| ⋄ w/o *False-Neg.* | 82.3 | 72.2 | 79.8 | 63.3 | 79.0 |
| ⋄ w/o LC | 80.0 | 70.7 | 80.4 | 65.1 | 78.6 |

Table 5: Ablation study on four components of QaDynamics. DC stands for Difficult Choice Selection, and LC stands for Low Confidence Selection. The following five columns denote the accuracy (%) on each benchmark.

| Data Selection | CommonsenseQA | | |
|---|---|---|---|
| | ID | SIQA (OOD) | PIQA (OOD) |
| Random 50% | 75.1 | 59.2 | 75.8 |
| Total data 100% | 78.0 | 60.4 | 75.8 |
| Easy-to-learn 50% | 71.2 | 58.1 | 75.4 |
| Ambiguous 50% | 76.4 | 61.1 | 76.5 |
| Hard-to-learn 50% | 76.2 | 60.6 | 76.4 |
| Difficult choice 60% | 77.7 | 60.6 | 76.9 |

Table 6: Supervised commonsense QA evaluation results on CSQA, SIQA and PIQA (Accuracy %). All experiments employ DeVERTa-v3-Base as the backbone.

## A.3 Ablation Study

In this section, we study the ablation of different components of our framework to determine the impact of using different data selection methods and strategies. There are four critical components that refine the QA dataset: Low Confidence (hard-to-learn) Selection, Difficult Choice Selection, *Mislabeled.*, and *False-Neg.*. The data selection details include adopting *Mislabeled.* removal and *False-Neg.* removal on the total data, selecting 50% of total data with the lowest confidence, and discarding the distractor with higher confidence.

To study the effect of different components, we train DeBERTa-v3-Large as the backbone by sequentially dropping the four components mentioned above one at a time. Their out-of-distribution performances on five different tasks are shown in Tab. 5. The results show that Difficult Choices Selection and Low Confidence Selection are effective strategies for improving the generalization ability of the model, and eliminating mislabeled examples and false negative distractors is also a useful approach for enhancing overall performance.

## A.4 Experiments on non-synthesized datasets

To assess the efficacy of our approach on non-synthesized datasets, we perform supplementary experiments using the training set of CommonsenseQA (Talmor et al., 2019). Subsequently, we evaluate the model on both the validation set of CommonsenseQA and other datasets such as SocialIQA (Sap et al., 2019b) and PIQA (Bisk et al., 2020). These datasets serve as in-domain and out-of-domain evaluations, respectively. The results are shown in Appx. §A.4. We observe that the *Hard-to-learn*, *Ambiguous*, and *Difficult choice* boost the out-of-domain performance compared to the baseline. It indicates that dropping easy choice con-

tributes to better generalization ability of the model. In the future, we may consider transferring such technique into more advanced but challenging commonsense tasks, such as commonsense knowledge base population (Fang et al., 2021b,a, 2023), causal reasoning (Chan et al., 2023; Wang et al., 2023c), and commonsense conceptualizations (Wang et al., 2023b,a; He et al., 2022).

## B Case Study

To further validate our framework, we present case studies in this section by showcasing instances selected by various strategies, which are illustrated in the Tab. 7.

***Mislabeled.* Removal.** Following Swayamdipta et al. (2020), we developed *Mislabeled.* to detect mislabeled examples, which involves extracting QA pairs with a ground answer that has low confidence. We have listed three mislabeled examples in the Tab. 7, where the confidence of the mislabeled option is relatively low compared to the other examples.

***False-Neg.* Removal.** Ma et al. (2021) mentioned the presence of false negative distractors that are also plausible in the QA pair. To address this issue, we implemented *False-Neg.* removal, which is designed to detect false negative distractors. We provide three examples of this strategy in action below. As shown in Tab. 7, the confidence of the false negative distractor is consistently close to 0.5, suggesting that its logit value is always in proximity to that of the correct choice during the training process, which is in line with the definition of *False-Neg.*. Moreover, based on these examples, we can infer that false negative distractors are often caused by insufficient content, resulting in multiple correct choices.

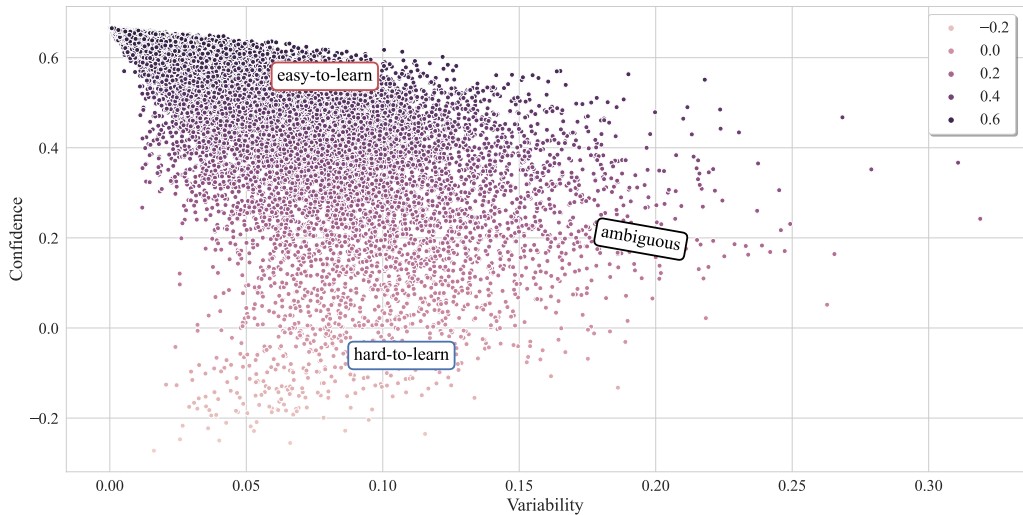

Figure 2: Demonstration of the synthetic QA dataset in the Zero-shot Commonsense QA task.

**Drop easy distractors.** Method *Difficult Choice* discards the option with higher confidence. To better understand the effectiveness of this method, we analyze the easy choice and identify two commonly occurring phenomena. First, we observed that the easier distractor is often more likely to contain grammatical errors, as demonstrated in the *Easy Distractors: Grammatical Error*. Second, the easy choice frequently has poor contextual relevance, as shown in the *Easy Distractors: Irrelevant to Context*, while these two phenomena can also be found in other instances shown in Tab. 7. Removing options that exhibit these features improves the overall quality of the dataset, which can lead to better performance and more reliable results.

| Question | Candidates | Confidence |
|---|---|---|
| **MISLABELED OPTION** | | |
| Flynn is cleaning out Flynn's garage. As a result, Flynn felt | A: confused. 
 B: elated. 
 C: scared. | A: 0.32 
 B: 0.22 
 C: 0.85 |
| Jamie decides to make some. Jamie is seen as | A: prepared. 
 B: knowledgeable 
 C: little. | A: 0.29 
 B: 0.76 
 C: 0.17 |
| Tracy can not wait to use it. As a result, Tracy wanted to | A: have fun. 
 B: do surveys. 
 C: eat. | A: 0.18 
 B: 0.32 
 C: 0.88 |
| **FALSE NEGATIVE DISTRACTORS** | | |
| Something that might happen as a consequence of watching a movie is | A: inspiration. 
 B: wood to burn. 
 C: being cultured | A: 1.00 
 B: 1.00 
 C: 0.51 |
| Ash gets Tracy's dog. As a result, Ash wanted to | A: writes exam. 
 B: leave. 
 C: make sure it is ok. | A: 0.99 
 B: 0.98 
 C: 0.51 |
| Bali maintains Ash relationship. Bali is seen as | A: committed. 
 B: generous. 
 C: adventourous. | A: 0.92 
 B: 0.51 
 C: 0.96 |
| **EASY DISTRACTORS: GRAMMATICAL ERROR** | | |
| Sydney takes the medicine. Sydney is seen as | A: reactive. 
 B: violence. 
 C: desperate. | A: 0.75 
 B: 0.98 
 C: 0.97 |
| an office building is for | A: advertising company. 
 B: purchase meals. 
 C: carrying the first class mail. | A: 0.99 
 B: 0.99 
 C: 0.67 |
| **EASY DISTRACTORS: IRRELEVANT TO CONTEXT** | | |
| You are likely to find a marker in | A: Afghanistanl. 
 B: school bag. 
 C: label | A: 0.99 
 B: 0.98 
 C: 0.77 |
| Pat opens the windows. As a result, others felt | A: snubbed 
 B: thankful 
 C: worried | A: 0.92 
 B: 0.88 
 C: 0.78 |

Table 7: Case studies of different strategies and easy choice. We highlight the golden standard , false negative , and easy distractors .