# OpenReview forum: "QADYNAMICS: Training Dynamics-Driven Synthetic QA Diagnostic for Zero-Shot Commonsense Question Answering"
_EMNLP/2023/Conference — EMNLP 2023 Findings_

### Official Review · Reviewer_k4z2 · 2023-07-20

**Soundness:** 3

**Excitement:**

3: Ambivalent: It has merits (e.g., it reports state-of-the-art results, the idea is nice), but there are key weaknesses (e.g., it describes incremental work), and it can significantly benefit from another round of revision. However, I won't object to accepting it if my co-reviewers champion it.

**Missing References:**

line 236, line 238, the references to the baseline models are missed.

Section 3.4, any references showing the human evaluation methods for QA-related datasets?

**Paper Topic And Main Contributions:**

This paper targets the problem of commonsense QA. While the current approaches to creating QA pairs for fine-tuning the PLMs are not of good quality, they propose a dynamics-driven framework to generate and select good QA pairs for commonsense fine-tuning.

**Questions For The Authors:**

Why specifically were 33% and 66% chosen for this experiment? Is there any ablation study showing that this is the best option besides adopting that from the baseline?

What is the difference between training dynamics and QA dynamics? Why there are no paragraphs talking about the training dynamics besides the baseline section?

How do you decide the number of choices for each datasets?



**Reasons To Accept:**

The overall methods are clearly stated: to create synthesised QA pairs and select part of them for PLM training, by implementation of QA dynamics of QA pairs compared with baseline training dynamics.

Overall the writing is concise and clear for a short paper.

The author also provides good results in terms of both metrics and human evaluations.

**Reasons To Reject:**

From line 45 to line 68, the challenges of this topic is discussed. However, it's hard to get any impression of the problems in this area without proper examples and explanations.

Section 2.2, the content of dynamics is not well explained. It's hard to tell the relationship between dynamics and logit scores, confidence and different checkpoints. Instead, the author only points out one or two references to the readers, which makes it hard to understand the motivations and idea of dynamics.

Results parts, we see good results as expected. However, there is no future discussion regarding these results. We are interested to know why the settings influence the overall results.

**Reproducibility:**

3: Could reproduce the results with some difficulty. The settings of parameters are underspecified or subjectively determined; the training/evaluation data are not widely available.

**Reviewer Confidence:**

3: Pretty sure, but there's a chance I missed something. Although I have a good feel for this area in general, I did not carefully check the paper's details, e.g., the math, experimental design, or novelty.

**Typos Grammar Style And Presentation Improvements:**

Table 1, repeated labels of "mislabelled" and "mislabeled removed". Repeated model size for training dynamic and QA dynamics.

In Table 2, the upper and down arrows indicate the best performance comes from the largest or smallest values (missing now).

Line 049, What is the triple, & tail? better explain it with KG

Line 194, what is very low confidence?

Formula 1, better useP(t_i|t1, t2, ..., t_{i-1}, t_{i+1}, ..., t_n)

Formula 2, why especially use T1 and A1 as an example, should try to use some general indication.

---

> ### Author Rebuttal · Authors · 2023-08-29
>
> Thanks for your dedicated review. Below please find our responses to your review.
>
> # Unclear Challenges
>
> Thank you for raising the issue of unclear explanation about challenging topics. The main three challenges are actually identified by Ma et al. (2021), who claim that:
>
> CSKB contains annotation artifacts that may bias the model from learning commonsense knowledge but rather seek easy backdoors to remember these artifacts.
> CSKBs also contain ungrammatical knowledge that is hard to be comprehended, and the synthesized QA pairs are hard to understand as well. For example, in ATOMIC, ``PersonX enjoys drinking, xEffect, stay up all night.``
> There is a lack of efficient negative sampling strategy to sample true-negative distractor at scale. For example, the triple in CSKB is ``PersonX feels hungry, xWant, eat a burger``. The sampled instance used for constructing distractors is ``PersonX likes Chinese food, xWant, eat dumplings``. In this case, the sampled distractor is also correct.
>
> Due to space constraints, there is not enough space for us to cover additional examples. We will add certain contents if we are granted another page for camera-ready.
>
> # Difference between QAdynamics and the original approach, and the content of QAdynamics
>
> Thank you for bringing up the issue regarding the content of dynamics. In the following paragraphs, we will first explain the content of QaDynamics, then illustrate the overlap and difference between QaDynamics and the original version and the motivation to propose these differences.
>
> In our approach, the logits are negative loss of each instance, ranging from $-\inf$ to $0$. The dynamics refer to the records of logits, obtained by evaluating the model on the whole training set after specific epochs or iterations. The confidence is computed via the dynamics recorded. Due to limited space, we illustrate the motivations and ideas of QaDynamics in the appendix.
>
> The **overlap** between the original version and QaDynamics is analysis towards data. They both obtain the logits of instances during training, compute the confidence and variability of data, and discriminate data into several areas. **QAdynamics different in the way that it is specifically designed for QA tasks.**
>
> ### Motivation for QADynamics
> In the original definition of confidence from Swayamdipta et al. (2020), it performs softmax operation on the logits. Nevertheless, **due to the larger number of distractors compared to the correct choice, the confidence of the correct choice tends to be underestimated.**
>
> For example, if there are five choices and the logits are -1, -3, -3, -3, -3, where the logit of ground truth is -1, the confidence of correct choice is 0.65 and the confidence of distractors is all 0.91 (marginal loss is used and the margin is set to be 1 in the original setting, so we assume the difference in logits of magnitude 2 is large enough).
>
> Furthermore, during the early stages of training, the model often provides random guesses and assigns equal probabilities to each choice. **This results in a lower confidence level for the correct choice compared to the distractors, leading to underestimation.**
>
> Additionally, **the presence of challenging choices or false negatives further contributes to the underestimation of the confidence in the original version.**
> We will add an figure illustrating that the confidence of the correct choice is consistently smaller than the confidence assigned to the distractors.
>
> ### Motivation for Data Selection
>
> Regarding data selection, when employing the original training dynamics, the confidence of an individual instance (comprising a single question with multiple choices) becomes a linear transformation of the confidence associated with the correct choice. **This transformation is primarily influenced by the difficulty measure attributed to the correct choice.**
>
>
> Consequently, the primary *motivation* behind our implementation is to **address the issue of underestimating the confidence in the correct choice, which arises from challenging or false negative distractors, as well as the imbalanced distribution between ground truth and distractors.** which is **the first difference.** This approach provides a more accurate measure towards the difficulty of each option, and derives the difficulty of instances according to each option, instead of only ground truth.
> The **second difference** is we adopt choice selection, which is a novel data selection method only applied to QA tasks.
> The **third difference** is our approach provides a method to detect false negative examples, which is a challenge caused by the QA synthesizing strategy.
>
>
> # Discussion towards results
>
> For the result parts, we mention the good performance of our approach, and conduct the ablation study by separating our approach into four parts. And we provide a direct analysis towards the results and the effect of each part by experiment results of ablation study, data annotation and the study on the confidence difference distribution. However, although there has been many works related to training dynamics (Aviad et al. 23, Fenia et al. 22, Tiberiu et al. 22), it is still unknown why hard-to-learn and ambiguous data is more effective for improving generalization ability. This remains as an interesting future work.
>
> # How to decide the amount of data
> For the quantity of data to be selected, the reason for choosing 66% is the training set has 3 options, and after dropping one distractor the data size will become 66% of original. To validate the effectiveness of choice selection, we set baselines with the same amount of data. Following Swayamdipta et al. (2020), we select 33% of total data. This measure is justifiable in previous works and ensures fair comparison.
>
> # How to decide number of choices
> The number of choices is actually a hyperparameter. As to the QA dataset proposed by Ma et al. (2020), it contains 2 distractors and we can only drop one of them (otherwise no distractor). For QA datasets which include more than 2 distractors, the number of distractors to be discarded is a hyperparameter decided before experiments, and additional experiments may be needed to select the number. However, there is no need to change our framework, as it is capable of dealing with arbitrary number of options in any QA dataset.
>
> # Missing references and representation improvements
>
> Thank you for mentioning the missing references and representation improvements. We will add the missing references and modify the unclear points in our paper. For the annotation, Wang et al. (2023) leverage the same annotation setup to evaluate the QA pairs’ selection results. We both aim for evaluating the effect of eliminating false-negative distractors and low-quality QA pairs, which we believe is a reliable measure.
>
> # References
>
> Wang, W., Fang, T., Ding, W., Xu, B., Liu, X., Song, Y., & Bosselut, A. (2023). CAR: Conceptualization-Augmented Reasoner for Zero-Shot Commonsense Question Answering. ArXiv, abs/2305.14869.
>
> Aviad Sar-Shalom and Roy Schwartz. 2023. Curating Datasets for Better Performance with Example Training Dynamics. In Findings of the Association for Computational Linguistics: ACL 2023, pages 10597–10608, Toronto, Canada. Association for Computational Linguistics.
>
> Swayamdipta, S., Schwartz, R., Lourie, N., Wang, Y., Hajishirzi, H., Smith, N.A., & Choi, Y. (2020). Dataset Cartography: Mapping and Diagnosing Datasets with Training Dynamics. Conference on Empirical Methods in Natural Language Processing.
>
> Ma, K., Ilievski, F., Francis, J.M., Bisk, Y., Nyberg, E., & Oltramari, A. (2020). Knowledge-driven Data Construction for Zero-shot Evaluation in Commonsense Question Answering. AAAI Conference on Artificial Intelligence.
>
> Fenia Christopoulou, Gerasimos Lampouras, and Ignacio Iacobacci. 2022. Training Dynamics for Curriculum Learning: A Study on Monolingual and Cross-lingual NLU. In Proceedings of the 2022 Conference on Empirical Methods in Natural Language Processing, pages 2595–2611, Abu Dhabi, United Arab Emirates. Association for Computational Linguistics.
>
> Tiberiu Sosea and Cornelia Caragea. 2022. Leveraging Training Dynamics and Self-Training for Text Classification. In Findings of the Association for Computational Linguistics: EMNLP 2022, pages 4750–4762, Abu Dhabi, United Arab Emirates. Association for Computational Linguistics.
>
> ***Thanks again for your valuable advice and we hope that our response will assist you in raising your score. If you need further clarification or discussion, please feel free to contact us.***

---

### Official Review · Reviewer_ukoU · 2023-07-27

**Soundness:** 3

**Excitement:**

3: Ambivalent: It has merits (e.g., it reports state-of-the-art results, the idea is nice), but there are key weaknesses (e.g., it describes incremental work), and it can significantly benefit from another round of revision. However, I won't object to accepting it if my co-reviewers champion it.

**Missing References:**

There is a previous paper that studied training dynamics on synthetic QA pairs from CSKBs, you should consider discussing it in your related work.

A Study of Zero-shot Adaptation with Commonsense Knowledge
J Zhang, F Ilievski, K Ma, J Francis, A Oltramari - Automated Knowledge Base Construction (AKBC), 2022

**Paper Topic And Main Contributions:**

This paper proposed a method for automatically detecting and filtering out noisy examples from synthetically constructed data from commonsense knowledge graphs. Inspired by the idea of datasets cartography (Swayamdipta et al. 20), the authors proposed to detect mislabeled examples, false-negative examples and easy-to-learn distractors from the synthetic data, using the confidence measure of the model along the training process. The final models are trained on the dataset after filtering. The authors evaluated the models on five commonsense reasoning datasets, and the results show that the proposed method greatly improved the performance compared to no filtering or filtering using the method from  Swayamdipta et al. 20.

**Questions For The Authors:**

1. In line 714, it should be resulting lower confidence in ground-truth answer right?
2. In line 722, if you’re using the second-lowest logits, then it should be the harder distractor?
3. I didn’t understand how is figure 1 computed? If you’re plotting the difference between the logits, what does softmax refer to? Your logits did not come from softmax?

**Reasons To Accept:**

The idea is very intuitive that filtering out noisy data from synthetic data potentially improves the model’s performance. The experimental results also show significant gains across five benchmarks by training on less but higher quality data.

**Reasons To Reject:**

Important technical details are not clearly explained, which weakens the claims made in this paper, in particular:
1. I don’t quite understand the motivation for Equation 2 and Equation 3. For equation 2, compare to Swayamdipta et al. 20, you just normalize over one distractor instead of all distractors. Intuitively, this just increased the probability of the correct answer and nothing more. For equation 3, if the correct answer has a high probability, the distractors will have low probabilities, thus these two measures (confidence for the correct answer and distractor) are strongly correlated or basically the same. Why don’t you use the original definition of confidence from Swayamdipta et al. 20? If you stick with the original training dynamics, and apply the easy distractor removal, mislabeled removal and false negative removal, would the model achieve similar performance?
2. For mislabeled, false-neg and mixed strategy, do you apply these methods on the remaining 33% of data to further filter out low-quality instances? If so, how much data do you filter out for each of the strategies? You should provide a table of statistics for this.

I am willing to increase my scores if my concerns and questions are addressed.

**Reproducibility:**

4: Could mostly reproduce the results, but there may be some variation because of sample variance or minor variations in their interpretation of the protocol or method.

**Reviewer Confidence:**

5: Positive that my evaluation is correct. I read the paper very carefully and I am very familiar with related work.

---

> ### Author Rebuttal · Authors · 2023-08-29
>
> Thanks for your dedicated review. Below please find our responses to your review.
>
> # Motivation of Our Implementation
>
> Thank you for addressing the concerns regarding the clarity of our implementation's motivation. We hereby clarify about the limitations of the original training dynamics approach in QA tasks and elucidate the rationale behind our proposed approach.
>
> ### Motivation for QADynamics
>
> First, the definition of logits in our task is the negative loss computed for each training instance, which is a real number, ranging from $-\inf$ to $0$, and larger logits indicate higher probability to be chosen.
> In the original definition of confidence from Swayamdipta et al. (2020), it performs softmax operation on the logits. Nevertheless, **due to the larger number of distractors compared to the correct choice, the confidence of the correct choice tends to be underestimated.**
>
> For example, if there are five choices and the logits are -1, -3, -3, -3, -3, where the logit of ground truth is -1, the confidence of correct choice is 0.65 and the confidence of distractors is all 0.91 (marginal loss is used and the margin is set to be 1 in the original setting, so we assume the difference in logits of magnitude 2 is large enough).
>
> Furthermore, during the early stages of training, the model often provides random guesses and assigns equal probabilities to each choice. **This results in a lower confidence level for the correct choice compared to the distractors, leading to underestimation.**
>
> Additionally, **the presence of challenging choices or false negatives further contributes to the underestimation of the confidence in the original version.**
> We will add an figure illustrating that the confidence of the correct choice is consistently smaller than the confidence assigned to the distractors.
>
> ### Motivation for Data Selection
>
> Regarding data selection, when employing the original training dynamics, the confidence of an individual instance (comprising a single question with multiple choices) becomes a linear transformation of the confidence associated with the correct choice. **This transformation is primarily influenced by the difficulty measure attributed to the correct choice.**
>
> Consequently, the primary *motivation* behind our implementation is to **address the issue of underestimating the confidence in the correct choice, which arises from challenging or false negative distractors, as well as the imbalanced distribution between ground truth and distractors.** Our approach aims to provide a more accurate assessment of the difficulty of choices or instances. We derive confidence of correct choice by comparing it with only one distractor, avoiding the imbalance of the amount of distractors. The selected distractor is the easier one, aiming to alleviate the impact of difficult or false negative distractors. In the figure 1 in Appendix, we prove that **our approach narrows the gap between confidence of correct choice and difficult choice, leading to a more justifiable learning results by our final QA model.**
>
> ### Experiment Justification
>
> |      Data selection Strategy      | Dynamics Version | aNLI | CSQA | PIQA | SIQA | WG   | Avg. |
> |-----------------------------------|------------------|------|------|------|------|------|------|
> | Hard-to-learn - Mixed Strategy    |     Original (Swayamdipta et al. 2020)     | 80.0 | 70.0 | 78.6 | 63.1 | 79.9 | 74.3 |
> | Hard-to-learn - Mixed Strategy    |    **QAdynamics**    | 80.4 | 71.3 | 79.8 | 64.9 | 78.9 | **75.1** |
> | Difficult Choice - Mixed Strategy |     Original (Swayamdipta et al. 2020)    | 81.2 | 70.5 | 81.1 | 63.4 | 80.0 | 75.2 |
> | Difficult Choice - Mixed Strategy |    **QAdynamics**    | 82.3 | 71.6 | 81.2 | 65.6 | 79.1 | **76.0** |
>
>
> Moreover, we conduct additional experiments to verify our approaches. The experiment results are shown in the table above.
>
> First, we discard mislabeled and false negative examples, and take 33% low-confidence data which is derived from the original version as training data. The average result of the original training dynamics is **74.3**, while the average result of our training dynamics is **75.1**, indicating our approach is more suitable for data selection on QA tasks.
>
> Then, we adopt the mixed strategy and perform choice selection to obtain 33% instances of total data based on original training dynamics as training data. The average result of the baseline is **75.2**, while the average performance of our training dynamics is **76.0** (we will add this result in ablation study).
>
> This shows firm proof that our option selection and QAdynamics are more useful than the original training dynamics in QA tasks.
>
> # Statistics of Each Strategy
>
> Thank you for raising the issue of lacking detailed statistics for each strategy. In our approach, we employ a threshold to filter out mislabeled and false negative data from the entire dataset. We select a portion of the data, **ensuring that the quantity of the selected data matches the quantity of the data before the dropping process**. This selected data is then used as our training dataset. As a result, there is no need to discard mislabeled and false negative data further. The rationale behind removing erroneous data from the overall dataset first is to **prioritize the assurance of data correctness before categorizing them into different difficulty levels.** Intuitively, it is essential to establish the accuracy and reliability of the data before proceeding with any further division or analysis. The threshold is decided based on rough observations of QAdynamic distributions, which emphasizes the balance between quantity and quality.
>
> Data statistics are shown in the table below.
>
> |                     |     Mislabeled     |     False Negative     |     Mixed Strategy     |     Data we used     |
> |----------|--------------|------|------|------|
> |   Data size   |   6465 (0.94%)   |       26320 (3.8%)     |      32875 (4.74%)    |          345775          |
>
>
> The data we utilized corresponds to the training data quantity during the training of the best models. The percentage figures presented in the table represent the proportion of erroneous data within the total dataset. As mentioned by Ma et al. (2021), the human accuracy on atomic and cwwv synthetic data is 78.0 and 80.7 respectively. The data discovered automatically by our strategy is **4.74% of total data, which is close to 25% of the poor-quality or grammatically wrong data**. And **most of them are located in the low-confidence areas, indicating our framework’s contribution towards purifying the low-quality data**.
>
> # Unclear points in our paper
>
> In line 714, it should be resulting lower confidence in ground-truth answer right?
> In line 722, if you’re using the second-lowest logits, then it should be the harder distractor?
> I didn’t understand how is figure 1 computed? If you’re plotting the difference between the logits, what does softmax refer to? Your logits did not come from softmax?
>
> Thank you for raising some unclear points in our paper.
> In question one, the confidence assigned to the ground-truth answers is consistently lower than the confidence assigned to the distractors. As previously mentioned, we will include graphs to visually illustrate this phenomenon.
>
> For question two, there is a typo in the logits expression. The logits should be negative loss. The higher the negative loss is, the corresponding choice is more likely to be chosen. Thus, it should be the lowest logit, which is the easier distractor.
>
> Figure 1 presents the difference between confidence of ground-truth answers and distractors. As mentioned above, our definition of logits is negative loss. This figure is derived by **random sampling the confidence of distractors and ground-truth answers and taking differences between them**. The blue part is the confidence difference derived from QAdynamics, while the orange part is from the original training dynamic. It indicates **our approach narrows the gap between confidence distribution of distractors and ground-truth answers**.
>
> We apologize for any typo that may have caused misunderstandings regarding our paper. To ensure the accuracy of our results and conclusions, we have thoroughly reviewed the experimental data and corresponding graphs. **We can confirm that the typos mentioned above do not impact the validity of our results and conclusions.**
>
> # Missing Reference
>
> Thank you for bringing the missing reference. We will add relavant discussions of this paper if we can get our paper accepted.
>
> # References
> Swayamdipta, S., Schwartz, R., Lourie, N., Wang, Y., Hajishirzi, H., Smith, N.A., & Choi, Y. (2020). Dataset Cartography: Mapping and Diagnosing Datasets with Training Dynamics. Conference on Empirical Methods in Natural Language Processing.
>
> Ma, K., Ilievski, F., Francis, J.M., Bisk, Y., Nyberg, E., & Oltramari, A. (2020). Knowledge-driven Data Construction for Zero-shot Evaluation in Commonsense Question Answering. AAAI Conference on Artificial Intelligence.
>
> ***Thanks again for your valuable advice and we hope that our response will assist you in raising your score. If you need further clarification or discussion, please feel free to contact us.***

---

### Official Review · Reviewer_3DE3 · 2023-08-04

**Typos Grammar Style And Presentation Improvements:** 1. The strategy expression in table 2…
**Soundness:** 3

**Excitement:**

3: Ambivalent: It has merits (e.g., it reports state-of-the-art results, the idea is nice), but there are key weaknesses (e.g., it describes incremental work), and it can significantly benefit from another round of revision. However, I won't object to accepting it if my co-reviewers champion it.

**Paper Topic And Main Contributions:**

This paper is targeted to reduce the noise in synthesized data for zero-shot commonsense question answering. It propose several strategies based on training dynamics to correct or filter out poor-quality data. According to the experiment results, these strategies can help the models achieve better average performance on five benchmarks.

**Questions For The Authors:**

Q1: The backbone model (RoBERTa-large) for derive QA Dynamics is different from the finally model for QA (DeBERTa-v3-large). Why?

Q2: Is the method also valuable for non-synthesized datasets?

**Reasons To Accept:**

The method is novel and the result is impressive. In comparison of original training dynamics, this work extends to multi-choice styled setting. The method proposed in this paper can improve the quality of synthesized commonsense QA data, and it seems to be also applicable for other QA tasks and datasets.

**Reasons To Reject:**

This paper does not report the extra time or space overhead. The QA Dynamics method requires to train an extra model and save its multiple checkpoints for the dataset diagnostic, which could be additional burden when applied to large-scale tasks. I wonder if larger models could be trained at the same cost and bring better performance.

**Reproducibility:**

3: Could reproduce the results with some difficulty. The settings of parameters are underspecified or subjectively determined; the training/evaluation data are not widely available.

**Reviewer Confidence:**

4: Quite sure. I tried to check the important points carefully. It's unlikely, though conceivable, that I missed something that should affect my ratings.

---

> ### Author Rebuttal · Authors · 2023-08-29
>
> Thanks for your dedicated review. Below please find our responses to your review.
>
> # Statistics of Extra Resources Needed
>
> Thank you for addressing the issue of extra resources required for obtaining dynamics. We believe that the additional resources invested in QaDynamics are manageable and justifiable.
>
> Firstly, it is important to note that the space and time resources needed for storing QAdynamics are relatively minimal compared to the dataset itself and conventional model training. The storage space required for storing QAdynamics is not substantial. To illustrate, our training set consists of 691,551 QA pairs, with each pair containing a question and three choices. Storing the QAdynamics of all QA pairs only necessitates an additional 16MB of storage. In total, 19 QA dynamic checkpoints are stored, resulting in a total requirement of $19*16=304$MB of additional storage. It is worth mentioning that this is the sole additional storage needed, as the model for generating QAdynamics computes on-the-fly and does not require local checkpoints. **Even for large-scale datasets, such as those containing 10 million QA pairs, a mere 4.4GB of additional storage is required. We believe this amount is insignificant in comparison to the storage required for the original datase**t.
>
> For the training time, indeed some time has to be invented to train a model for generating the QAdynamics. In our case, we dedicated a total of 56.5 hours to obtain the training dynamics by training a Roberta-large model. Nevertheless, training the final QA model becomes less time-consuming due to the reduced quantity of data after the data selection process. The selected data can also be helpful in saving time during future training endeavors. Thus, **the overall training time is not extensively expanded**.
>
> # Model Selection for Computing QAdynamics
>
> Thank you for addressing this matter. Our choice to utilize the RoBERTa-Large model for computing QAdynamics is grounded in a couple of significant factors.
>
> Firstly, the RoBERTa-Large model possesses a smaller size compared to DeBERTa-Large, **resulting in faster training and inference speeds**. Given that QAdynamics necessitates multiple on-the-fly inferences, opting for RoBERTa-Large is a more practical and feasible option.
>
> Secondly, through our testing, we have observed that RoBERTa-Large, when combined with choice selection and mixed strategy, **achieves a noteworthy performance average of **68.1** across five QA benchmarks. This performance surpasses the results reported by Ma et al. (2021)**. Thus, we have confidence in RoBERTa-Large's ability to effectively learn the QA set and differentiate between easy and difficult QA pairs.
>
> Lastly, it is important to consider the inherent challenge in expecting a model to learn from data that is itself difficult to learn. If we were to exclusively select difficult-to-learn samples for the model and expect it to learn from them, it could potentially lead to further confusion. Taking all these factors into account, we have made the decision to employ RoBERTa-Large as the QAdynamics model.
>
> # Performance on non-synthesized datasets
>
> Thank you for bringing up the concern regarding non-synthesized datasets. To address this, we conducted additional experiments where we trained the model using the **training set of CommonsenseQA and evaluated it on the dev sets of CommonsenseQA (in domain), SocialIQA (out-of-domain), and PIQA (out-of-domain)**. During the training process, we employed various QADynamics strategies, and the outcomes are presented in the following results.
>
> |     Data Type           | CommonsenseQA (In Distribution) |    SocialIQA (OOD)    |    PIQA (OOD)    |
> |---------------------------|--------------------|--------------------|--------------|
> |     Random              |        75.1         |        60.1         |     75.9     |
> |     Total data            |        78.0         |        60.4         |     75.8     |
> |     easy-to-learn       |        71.2         |        58.1         |     75.4     |
> |     ambiguous          |        76.4         |        61.1         |     76.5     |
> |     hard-to-learn       |        76.2         |        60.6         |     76.4     |
> |     difficult choice     |        77.7         |        60.6         |     76.9     |
>
> Based on the results, we have observed the effectiveness of QADynamics on non-synthesized QA benchmarks, such as CommonsenseQA. Furthermore, it enhances the generalizability of the QA model to other commonsense QA benchmarks. Therefore, we strongly believe that **our proposed method is not only advantageous for zero-shot commonsense QA but also invaluable for supervised QA models**.
>
> # Missing References and Naming Misunderstanding
>
>
> Thank you for bringing the writing issues of our paper to our attention. We appreciate your feedback and are dedicated to improving the clarity and coherence of our work.
>
> With regard to Table 2, we acknowledge that the term "After Mislabeled" may be a little misleading. In the camera ready version, we will make the necessary revisions to provide a more descriptive label, such as "After Removing Mislabeled," to accurately reflect the process.
>
> Due to space limitations, we were unable to provide extensive details about the baselines and datasets. However, if granted an additional page in the camera ready version, we are fully committed to including all the necessary information. Specifically, the baselines referred to as "Total 100% (Ma et al., 2021)" employ DeBERTa-v3-Large as the backbone. It is important to note that all experiments presented in Table 1 are based on DeBERTa-v3-Large for fair comparisons.
>
> We assure you that in the final version of the paper, we will address the missing reference and ensure that all abbreviations are clearly explained.
>
> # References
> Swayamdipta, S., Schwartz, R., Lourie, N., Wang, Y., Hajishirzi, H., Smith, N.A., & Choi, Y. (2020). Dataset Cartography: Mapping and Diagnosing Datasets with Training Dynamics. Conference on Empirical Methods in Natural Language Processing.
>
> Ma, K., Ilievski, F., Francis, J.M., Bisk, Y., Nyberg, E., & Oltramari, A. (2020). Knowledge-driven Data Construction for Zero-shot Evaluation in Commonsense Question Answering. AAAI Conference on Artificial Intelligence.
>
> ***Thanks again for your valuable advice and we hope that our response will assist you in raising your score. If you need further clarification or discussion, please feel free to contact us.***

---

### Meta-Review · Area_Chair_B31f · 2023-09-18

**Recommendation:** 4

**Metareview:**

**Pros**: the methods is novel and result is impressive as all reviewers agreed.

**Cons**: there are confusions or understanding issues raised by multiple reviewers. They all grasp the confusing parts after author discussion though. Hope the authors could include the explanations in the updated version of the paper.

Overall, This short paper proposed a novel framework with training dynamics to filter low-quality data for multi-choice selection problem. The results is impressive compared to multiple baselines.

---

### Decision · Program_Chairs · 2023-10-07

**Decision:**

Accept-Findings

**Comment:**

**Pros**: the methods is novel and result is impressive as all reviewers agreed.

**Cons**: there are confusions or understanding issues raised by multiple reviewers. They all grasp the confusing parts after author discussion though. Hope the authors could include the explanations in the updated version of the paper.

Overall, This short paper proposed a novel framework with training dynamics to filter low-quality data for multi-choice selection problem. The results is impressive compared to multiple baselines.